# Unsupervised Domain Adaptation for Distance Metric Learning

**Kihyuk Sohn**[1]    **Wenling Shang**[2]    **Xiang Yu**[1]    **Manmohan Chandraker**[1,3]
[1]NEC Labs America    [2]University of Amsterdam    [3]UC San Diego

## Abstract

Unsupervised domain adaptation is a promising avenue to enhance the performance of deep neural networks on a target domain, using labels only from a source domain. However, the two predominant methods, domain discrepancy reduction learning and semi-supervised learning, are not readily applicable when source and target domains do not share a common label space. This paper addresses the above scenario by learning a representation space that retains discriminative power on both the (labeled) source and (unlabeled) target domains while keeping representations for the two domains well-separated. Inspired by a theoretical analysis, we first reformulate the disjoint classification task, where the source and target domains correspond to *non-overlapping class labels*, to a verification one. To handle both within and cross domain verifications, we propose a *Feature Transfer Network* (FTN) to separate the target feature space from the original source space while aligned with a *transformed* source space. Moreover, we present a non-parametric *multi-class entropy minimization* loss to further boost the discriminative power of FTNs on the target domain. In experiments, we first illustrate how FTN works in a controlled setting of adapting from MNIST-M to MNIST with disjoint digit classes between the two domains and then demonstrate the effectiveness of FTNs through state-of-the-art performances on a cross-ethnicity face recognition problem.

## 1 Introduction

Despite strong performances on facial analysis using deep neural networks (Taigman et al., 2014; Sun et al., 2014; Schroff et al., 2015; Parkhi et al., 2015), learning a model that generalizes across variations in attributes like ethnicity, gender or age remains a challenge. For example, it is reported by Buolamwini & Gebru (2018) that commercial engines tend to make mistakes at detecting gender for images of darker-skinned females. Such biases have enormous social consequences, such as conscious or unconscious discrimination in law enforcement, surveillance or security (WIRED, 2018a;b; NYTimes, 2018; GIZMODO, 2018). A typical solution is to collect and annotate more data along the underrepresented dimension, but such efforts are laborious and time consuming. This paper proposes a novel deep unsupervised domain adaptation approach to overcome such biases in face verification and identification.

Deep domain adaptation (Long et al., 2013; 2015; 2016; Tzeng et al., 2015; Ganin et al., 2016; Sohn et al., 2017; Haeusser et al., 2017; Luo et al., 2017) allows porting a deep neural network to a target domain without extensive labeling efforts. Currently, there are two predominant approaches to deep domain adaptation. The first approach, domain divergence reduction learning, is motivated by the works of (Ben-David et al., 2007; 2010). It aims to reduce the source-target domain divergence using domain adversarial training (Ganin et al., 2016; Sohn et al., 2017; Tran et al., 2018) or maximum mean discrepancy minimization (Tzeng et al., 2015; Long et al., 2015; 2016), while leveraging supervised loss from labeled source examples to maintain feature space discriminative power. Since the theoretical basis of this approach (Ben-David et al., 2007) assumes a common task between domains, it is usually applied to a classification problem where the source and target domains share the same label space and task definition. The second approach considers domain adaptation as a semi-supervised learning problem and applies techniques such as entropy minimization (Grandvalet & Bengio, 2005) or self-ensembling (Laine & Aila, 2017; Tarvainen & Valpola, 2017; French et al., 2018) on target examples to encourage decisive and consistent predictions.

However, neither of those are applicable if the label spaces of source and target domains do not align. As a motivating example, consider a cross-ethnicity generalization of face recognition problem,

where the source ethnicity (e.g., Caucasian) contains labeled examples and the target ethnicity (e.g., African-American) contains only unlabeled examples. When it is cast as a classification problem, the tasks of the two domains are different due to disjoint label spaces. Moreover, examples from different ethnicity domains almost certainly belong to different identity classes. To satisfy such additional label constraints, representations of examples from different domains should ideally be distant from each other in the embedding space, which conflicts with the requirements of domain divergence reduction learning as well as entropy minimization on target examples with source domain class labels.

In this work, we aim at learning a shared representation space between a source and target domain with disjoint label spaces that not only remains discriminative over both domains but also keep representations of examples from different domains well-separated, when provided with additional label constraints. Firstly, to overcome the limitation of domain adversarial neural network (DANN) (Ganin et al., 2016), we propose to convert disjoint classification tasks (i.e., the source and target domains correspond to non-overlapping class labels) into a unified binary verification task. We term adaptation across such source and target domains as *cross-domain distance metric adaptation (CD2MA)*. We demonstrate a generalization of the theory of domain adaptation (Ben-David et al., 2007) to our setup, which bounds the empirical risk for within-domain verification of two examples drawn from the unlabeled target domain. While the theory does not guarantee verification between examples from different domains, we propose approaches that also address such cross-domain verification tasks.

To this end, we introduce a *Feature Transfer Network* (FTN) that separates the target features from the source features while simultaneously aligning them with an auxiliary domain of transformed source features. Specifically, we learn a shared feature extractor that maps examples from different domains to representations far apart. Simultaneously, we learn a *feature transfer module* that transforms the source representation space to another space used to align with the target representation space through a domain adversarial loss. By forging this alignment, the discriminative power from the augmented source representation space would ideally be transferred to the target representation space. The verification setup also allows us to introduce a novel entropy minimization loss in the form of $N$-pair metric loss (Sohn, 2016), termed *multi-class entropy minimization* (MCEM), to further leverage unlabeled target examples whose label structure is not known. MCEM samples pairs of examples from a discovered label structure within the target domain using an offline hierarchical clustering algorithm such as HDBSCAN (Campello et al., 2013), computes the $N$-pair metric loss among these examples (Sohn, 2016), and backpropagates the resulting error derivatives.

In experiments, we first perform on a controlled setting by adapting between disjoint sets of digit classes. Specifically, we adapt from 0–4 of MNIST-M (Ganin et al., 2016) dataset to 5–9 of MNIST dataset and demonstrate the effectiveness of FTN in learning to align and separate domains. Then, we assess the impact of our proposed unsupervised CD2MA method on a challenging cross-ethnicity face recognition task, whose source domain contains face images of Caucasian identities and the target domain of non-Caucasian identities, such as African-American or East-Asian. This is an important problem since existing face recognition datasets show significant label biases towards Caucasian ethnicity, leading to sub-optimal recognition performance for other ethnicities. The proposed method demonstrates significant improvement in face verification and identification compared to a source-only baseline model and a standard DANN. Our proposed method also closely matches the performance upper bounds obtained by training with fully labeled source and target domains.

## 2 RELATED WORK

Research efforts in deep domain adaptation have explored a proper metric to measure the variational distance between two domains and subsequently regularize neural networks to minimize this distance. For example, maximum mean discrepancy (Long et al., 2013; 2016; Tzeng et al., 2014; Fernando et al., 2015; Tzeng et al., 2015; Sun & Saenko, 2016) estimates the domain difference based on kernels. As another example, domain adversarial neural networks (Ganin et al., 2016; Bousmalis et al., 2016; 2017; Sohn et al., 2017; Luo et al., 2017; Tran et al., 2018), measuring the distance using a trainable and flexible discriminator often parameterized by an MLP, have been successfully adopted for several computer vision applications, such as semantic segmentation (Hoffman et al., 2016; Tsai et al., 2018; Zhang et al., 2018) and object detection (Chen et al., 2018). Most of those works assume a common classification task between two domains, whereas we tackle a cross-domain distance metric adaptation problem where label spaces of source and target domains are different.

Moreover, our problem setting, an adaptation from labeled source to unlabeled target with disjoint label spaces, contains flavors from both domain adaptation (DA) and transfer learning (TL), following

the nomenclature of (Pan et al., 2010). The difference in input distribution between source and target domains and the lack of labels in the target domain are similar to that of DA or transductive TL (Pan et al., 2010), while the difference in label distribution and task definitions between two domains is akin to inductive TL (Pan et al., 2010; Daumé III, 2007). In our work, we formalize this problem in domain adaptation framework using verification as a common task. This is a key contribution that allows theoretical analysis on the generalization bound as presented in Section 3 and Appendix A, while allowing novel applications like cross-ethnicity face recognition.

In terms of task objective, (Hu et al., 2015; Ganin et al., 2016; Sohn et al., 2017) also deal with domain adaptation in distance metric learning, but neither learns a representation space capable of separating the source and target domains. Resembling CD2MA, Luo et al. (2017) considers domain adaptation with disjoint label spaces, but the problem is still cast as classification with an assumption that the target label space is known and a few labeled target examples are provided for training.

In terms of network design, residual transfer network (Long et al., 2016), which learns two classifiers differ by a residual function for the source and the target domain, is closely related. However, it only tackles the scenario where source and target domains share a common label space for classification.

## 3 REVISITING THE THEORY OF DOMAIN ADAPTATION FOR VERIFICATION

Under the domain adaptation assumption, Ben-David et al. (2007) show that the empirical risk on the target domain $\mathcal{X}_T$ is bounded by the empirical risk on the source domain $\mathcal{X}_S$ and the variational distance between the two domains, provided that the source and the target domains share the classifiers. Therefore, this bound is not applicable to our CD2MA setup where the label spaces of two domains are often different. To generalize those theoretical results to our setting, we reformulate the verification task as a binary classification task shared across two domains. This new binary classification task takes a pair of images as an input and predicts the label of 1 if the pair of images shares the same identity and 0 otherwise. Furthermore, if we now define the new source domain to be pairs of source images and the new target domain to be pairs of target images, then Theorem 1 and 2 from (Ben-David et al., 2007) can be directly carried over to bound the new target domain binary classification error in the same manner. That is, the empirical with-in target domain verification loss is bounded by with-in source domain verification loss and the variational distance between $\mathcal{X}_S \times \mathcal{X}_S$ and $\mathcal{X}_T \times \mathcal{X}_T$.[1] Note that inputs to the binary classifier are pairs of images from the same domain. Thus, this setup only addresses adaptation of within-domain verification to unlabeled target domains.

There are two implications from the theoretical insights on domain adaptation using verification as a shared classification task. Firstly, domain adversarial training, reducing the discrepancy between the source and the target product spaces, coupled with supervised source domain binary classification loss (i.e., verification loss using source domain labels) can yield target representations with high discriminative power when performing within-domain verification. Note that in practice we approximately reduce the product space discrepancy by generic adversarial learning as done in (Ganin et al., 2016; Sohn et al., 2017). Secondly, there is no guarantee that the aligned source and target feature spaces possess any discriminative power for cross-domain verification task. Thus, additional actions in the form of a feature transfer module and domain separation objective are required to address this issue. These two consequences together motivate the design of our proposed framework, which is introduced in the next section.

## 4 FEATURE TRANSFER NET: LEARNING TO ALIGN AND SEPARATE DOMAINS

In this section, we first define the CD2MA problem setup and motivate our proposed feature transfer network (FTN). Then we elaborate on the training objectives that help our model achieve its desired properties. Lastly, we provide practical considerations to implement our proposed algorithm.

### 4.1 PROBLEM STATEMENT AND ALGORITHM OVERVIEW

Recall the description of CD2MA, given source and target domain data distributions $\mathcal{X}_S$ and $\mathcal{X}_T$, our goal is to verify whether two random samples $x, x'$ drawn from either of the two distributions (and we do not know which distribution $x$ or $x'$ come from a priori) belong to the same class.

There are 3 scenarios of constructing a pair: $x, x' \in \mathcal{X}_S$, $x, x' \in \mathcal{X}_T$, or $x \in \mathcal{X}_S, x' \in \mathcal{X}_T$. We refer the task of the first two cases as *within-domain* verification and the last as *cross-domain* verification.

---

[1]Mathematical details formalizing our theoretical analysis are in the Appendix A.

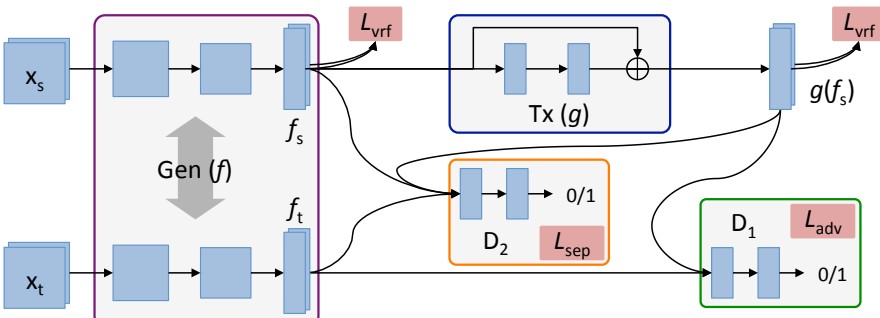

**Figure 1:** Training of Feature Transfer Network (FTN) for verification, composed of feature generation module (Gen; $f$), feature transfer module (Tx; $g$), and two domain discriminators $D_1$ and $D_2$. Verification objective $\mathcal{L}_{\mathrm{vrf}}$'s are applied to source ($f_s$) pairs and transformed source ($g(f_s)$)) pairs. Our FTN applies domain adversarial objective $\mathcal{L}_{\mathrm{adv}}$ for domain alignment between transformed source and target domains by $D_1$ and applies $\mathcal{L}_{\mathrm{sep}}$ to distinguish source domain from both target and transformed source domains by $D_2$.

If $x, x' \in \mathcal{X}_S$ (or $\mathcal{X}_T$), we need a source (or target) domain classifier[2]. For the source domain, we are provided with adequate labeled training examples to learn a competent classifier. For the target domain, we are only given unlabeled examples. However, with our extension of Theorem 1 and 2 from (Ben-David et al., 2007), discriminative power of the classifier can be transferred to the target domain by adapting the representation spaces of $\mathcal{X}_T \times \mathcal{X}_T$ and $\mathcal{X}_S \times \mathcal{X}_S$, that is, we can utilize the same competent classifier from the source domain to verify target domain pairs if two domains are well-aligned. For the third scenario where $x \in X_S$ but $x' \in X_T$, we assume that the two examples *cannot* be of the same class, which is true for problems such as cross-ethnicity face verification.

Our proposed framework, *Feature Transfer Network* (FTN), is designed to solve all these verification scenarios in an unified framework. FTN is composed of multiple modules as illustrated in Figure 1. First, a feature generation module $f : \mathcal{X} \to \mathcal{Z}$ denoted as "Gen" in Figure 1 ideally maps $\mathcal{X}_S$ and $\mathcal{X}_T$ to distinguishable representation spaces, that is, $f(\mathcal{X}_S)$ and $f(\mathcal{X}_T)$ are far apart. To achieve this, we introduce a *domain separation objective*.[3] Next, the feature transfer module $g : \mathcal{Z} \to \mathcal{Z}$ denoted as "Tx" in Figure 1 transforms $f(\mathcal{X}_S)$ to $g(f(\mathcal{X}_S))$ for it to be aligned with $f(\mathcal{X}_T)$. To achieve this, we introduce a *domain adversarial objective*. Finally, we apply verification losses on $f(\mathcal{X}_S)$ and $g(f(\mathcal{X}_S))$ using classifiers $h_f, h_g : \mathcal{Z} \times \mathcal{Z} \to \{0, 1\}$. During testing, we compare the metric distance between $f(x)$ and $f(x')$. Overall, we achieve the following desired capabilities:

- If $x, x'$ are from different domains, $f(x)$ and $f(x')$ will be far away due to the functionality of the feature generation module.
- If $x, x' \in \mathcal{X}_S$, then $f(x)$ and $f(x')$ will be close if they belong to the same class and far away otherwise, due to the discriminative power acquired from optimizing $h_f$.
- If $x, x' \in \mathcal{X}_T$, then $f(x)$ and $f(x')$ will be close if they belong to the same class and far otherwise, due to the discriminative power acquired by optimizing $h_g$ with domain adversarial training.

## 4.2 TRAINING OBJECTIVES

We first define individual learning objectives of the proposed Feature Transfer Network and then present overall training objectives of FTN. For ease of exposition, all objectives are to be maximized.

**Verification Objective.** For a pair of source examples, we evaluate the verification losses at two representations spaces $f(\mathcal{X}_S)$ and $g(f(\mathcal{X}_S))$ using classifiers $h_f$ and $h_g$ as follows:

$$\mathcal{L}_{\mathrm{vrf}}(f) = \mathbb{E}_{(x_1,x_2)\in\mathcal{X}_S\times\mathcal{X}_S}\big[y_{12}\log h_f(f_1, f_2) + (1-y_{12})\log(1-h_f(f_1, f_2))\big] \quad (1)$$

$$\mathcal{L}_{\mathrm{vrf}}(g) = \mathbb{E}_{(x_1,x_2)\in\mathcal{X}_S\times\mathcal{X}_S}\big[y_{12}\log h_g(g_1, g_2) + (1-y_{12})\log(1-h_g(g_1, g_2))\big] \quad (2)$$

where $g_i = g(f(x_i))$, $f_i = f(x_i)$ and $y_{12} = 1$ if $x_1$ and $x_2$ are from the same class and 0 otherwise. While classifiers $h_f, h_g$ can be parameterized by neural networks, we aim to learn a generator $f$ and $g$ whose embeddings can be directly used as a distance metric. Therefore, we use non-parameteric classifiers $h_f = \sigma(f_1^\top f_2)$, $h_g = \sigma(g_1^\top g_2)$ where $\sigma(a) = \frac{1}{1+\exp(-a)}$.

---

[2]Here we use the term "classifier" to denote a prediction module for the verification of a pair.

[3]The term "domain separation" indicates that the representation space can be separated with respect to domain definitions (such as, source or target). This is unrelated to Domain Separation Network (Bousmalis et al., 2016), where it denotes the separation of the representation space into shared and private subspaces.

**Domain Adversarial Objective.** Let $D_1 : \mathcal{Z} \to (0, 1)$ be a domain discriminator. As mentioned earlier, $D_1$ is trained to discriminate distributions $f(\mathcal{X}_T)$ and $g(f(\mathcal{X}_S))$ and then produces gradient for them to be indistinguishable. The learning objectives are written as follows:

$$\mathcal{L}_{D_1} = \mathbb{E}_{x \in \mathcal{X}_S} \log D_1(g) + \mathbb{E}_{x \in \mathcal{X}_T} \log\left(1 - D_1(f)\right), \; \mathcal{L}_{\text{adv}} = \mathbb{E}_{x \in \mathcal{X}_T} \log D_1(f) \qquad (3)$$

Note that when feature transform module is an identity mapping, i.e., $g(f(x)) = f(x)$, Equation (3) defines the training objective of standard DANN.

**Domain Separation Objective.** The goal of this objective is to distinguish between source and target at representation spaces of generation module. To this end, we formulate the objective using another domain discriminator $D_2 : \mathcal{Z} \to (0, 1)$:

$$\mathcal{L}_{\text{sep}} = \mathbb{E}_{x \in \mathcal{X}_S} \log D_2(f) + \frac{1}{2}\left[\mathbb{E}_{x \in \mathcal{X}_S} \log(1 - D_2(g)) + \mathbb{E}_{x \in \mathcal{X}_T} \log(1 - D_2(f))\right] \qquad (4)$$

Note that, in $\mathcal{L}_{\text{sep}}$, the source space $f(\mathcal{X}_S)$ is not only pushed apart from the target space $f(\mathcal{X}_T)$ but also from the augmented source space $g(f(\mathcal{X}_S))$ to ensure that $g$ learns meaningful transformation of source domain representation beyond identity transformation.

**Training FTN.** Now we are ready to present the overall training objectives $\mathcal{L}_f$ and $\mathcal{L}_g$:

$$\mathcal{L}_f = \frac{1}{2}\left[\mathcal{L}_{\text{vrf}}(g) + \mathcal{L}_{\text{vrf}}(f)\right] + \lambda_1 \mathcal{L}_{\text{adv}} + \lambda_2 \mathcal{L}_{\text{sep}}, \; \mathcal{L}_g = \mathcal{L}_{\text{vrf}}(g) + \lambda_2 \mathbb{E}_{\mathcal{X}_S} \log(1 - D_2(g)) \qquad (5)$$

with $\lambda_1$ for domain adversarial objective and $\lambda_2$ for domain separation objective. We use $\mathcal{L}_{D_1}$ in Equation (3) for $D_1$ and $\mathcal{L}_{D_2} = \mathcal{L}_{\text{sep}}$ for $D_2$. We alternate updating between $D_1$ and $(f, g, D_2)$.

## 4.3 PRACTICAL CONSIDERATIONS

**Preventing Mode Collapse via Feature Reconstruction Loss.** The mode collapsing phenomenon with generative adversarial networks (GANs) (Goodfellow et al., 2014) has received much attention (Salimans et al., 2016). In the context of domain adaptation, we also find it critical to treat the domain adversarial objective with care to avoid similar optimization instability.

In this work, we prevent the mode collapse issue for domain adversarial learning with an additional regularization method similar to (Sohn et al., 2017). Assuming the representation of the source domain is already close to optimal, we regularize the features of source examples to be similar to those from the reference network $f_{\text{ref}} : \mathcal{X} \to \mathcal{Z}$, which is pretrained on labeled source data and fixed during the training of $f$. Furthermore, we add a similar but less emphasized ($\lambda_4 < \lambda_3$) regularization to target examples, simultaneously avoiding collapsing and allowing more room for target features to diverge from the original representations. Finally, the feature reconstruction loss is written as follows:

$$\mathcal{L}_{\text{recon}} = -\left[\lambda_3 \mathbb{E}_{x \in \mathcal{X}_S} \|f(x) - f_{\text{ref}}(x)\|_2^2 + \lambda_4 \mathbb{E}_{x \in \mathcal{X}_T} \|f(x) - f_{\text{ref}}(x)\|_2^2\right] \qquad (6)$$

We empirically find that without the feature reconstruction loss, the training would become unstable, reach an early local optimum and lead to suboptimal performance (see Section 6 and Appendix C). Thus, we always include the feature reconstruction loss to train DANN or FTN models unless stated otherwise.

**Replacing Verification Loss with $N$-pair Loss.** Our theoretical analysis in Section 3 (and Appendix A) suggests to use a verification loss that compares similarity between a pair of images. In practice, however, the pairwise verification loss is too weak to learn a good deep distance metric. Following (Sohn, 2016), we propose to replace the verification loss with an $N$-pair loss, defined as follows:

$$\mathcal{L}_N(f) = \mathbb{E}_{\{x_n, x_n^+\}_{n=1}^N, x_n, x_n^+ \in \mathcal{X}_S}\left[\sum_{n=1}^N \log p_n(f)\right], \; p_n(f) = \frac{\exp(f(x_n)^\top f(x_n^+))}{\sum_{k=1}^N \exp(f(x_n)^\top f(x_k^+))} \qquad (7)$$

where $x_n$ and $x_n^+$ are from the same class and $x_n$ and $x_k^+$, $n \neq k$, are from different classes. Replacing $\mathcal{L}_{\text{vrf}}$ into $\mathcal{L}_N$, the training objective of FTN with $N$-pair loss is written as follows:

$$\mathcal{L}_f = \frac{1}{2}\left[\mathcal{L}_N(g) + \mathcal{L}_N(f)\right] + \lambda_1 \mathcal{L}_{\text{adv}} + \lambda_2 \mathcal{L}_{\text{sep}} + \mathcal{L}_{\text{recon}}, \; \mathcal{L}_g = \mathcal{L}_N(g) + \lambda_2 \mathbb{E}_{\mathcal{X}_S} \log(1 - D_2(g)) \quad (8)$$

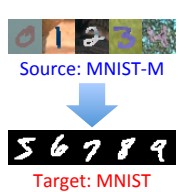 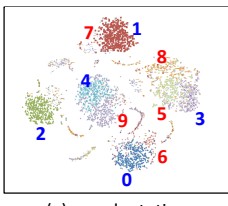 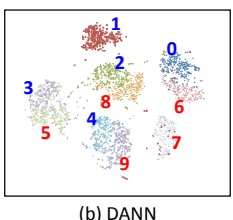 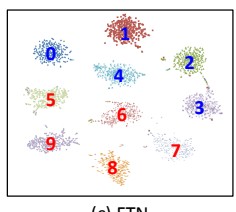

Source: MNIST-M

Target: MNIST

(a) no adaptation      (b) DANN      (c) FTN

**Figure 2:** t-SNE visualizations of source (0–4 from MNIST-M) and target (5–9 from MNIST) representations by different learning methods: (a) deep neural network without adaptation, (b) domain adversarial neural network (DANN) and (c) our feature transfer network (FTN). While domain adversarial learning results in significant confusion of digits classes between source and target domains (e.g., 3/5, 2/8, 4/9, or 0/6 in (b)), the proposed FTN transfers discriminative power to target domain while successfully separating them from the source domain.

## 5   ENTROPY MINIMIZATION VIA HIERARCHICAL CLUSTERING

Entropy minimization (Grandvalet & Bengio, 2005) is a popular training objective in unsupervised domain adaptation: unlabeled data is trained to minimize entropy of a class prediction distribution so as to form features that convey confident decision rules. However, it is less straightforward how to apply entropy minimization when label spaces for source and target are disjoint. Motivated from Section 3, we extend entropy minimization for distance metric adaptation using verification as a common task for both domains:

$$\mathcal{L}_{\text{vrf}}^{\text{ent}}(f) = \mathbb{E}_{x_i, x_j \in \mathcal{X}_T} \big[ p_{ij} \log p_{ij} + (1 - p_{ij}) \log(1 - p_{ij}) \big] \tag{9}$$

where $p_{ij} \triangleq p_{ij}(f) = \sigma(f(x_i)^\top f(x_j))$. This formulation encourages a more confident prediction for verifying two unlabeled images, whether or not coming from the same class.

However, recall that for the source domain, we use $N$-pair loss instead of pair-wise verification loss for better representation learning. Therefore, we would like to similarly incorporate the concept of $N$-pair loss on the target domain by forging a multi-class entropy minimization (MCEM) objective. This demands $N$ pair examples to be sampled from the target domain. As the target domain is unlabeled, we ought to first discover a plausible label structure, which is done off-line via HDBSCAN (Campello et al., 2013; McInnes et al., 2017), a fast and scalable density-based hierarchical clustering algorithm. The returned clusters provide pseudo-labels to individual examples of the target domain, allowing us to sample $N$ pair examples to evaluate the following MCEM objective:

$$\mathcal{L}_N^{\text{ent}}(f) = \mathbb{E}_{\{x_n, x_n^+\}_{n=1}^N, x_n, x_n^+ \in \mathcal{X}_T} \Big[ \sum_{n=1}^N \big\{ \sum_{m=1}^N p_{nm} \log p_{nm} \big\} \Big], \; p_{nm}(f) = \frac{\exp(f_n^\top f_m^+)}{\sum_{k=1}^N \exp(f_n^\top f_k^+)} \tag{10}$$

where $x_n$ and $x_n^+$ are from the same cluster and $x_n$ and $x_k^+$, $n \neq k$ are from different clusters. The objective can be combined with $\mathcal{L}_f$ in Equation (8) to optimize $f$.

## 6   EXPERIMENTS

In this section, we first experiment on digit datasets as a proof of concept and compare our proposed FTN to DANN. Then, we tackle the problem of cross-ethnicity generalization in the context of face recognition to demonstrate the effectiveness of FTN. In all experiments, we use $N$-pair loss as defined in Equation (8) to update $f$ and $g$ for better convergence and improved performance. We also use the same learning objectives for DANN while fixing $g$ to the identity mapping and $\lambda_2 = 0$.

### 6.1   PROOF OF CONCEPT: MNIST-M (0–4) TO MNIST (5–9)

To provide insights on the functionality of FTN, we conduct an experiment adapting the digits 0–4 from MNIST-M (Ganin et al., 2016) to 5–9 from MNIST. In other words, the two domains in our setting not only differ in foreground and background patterns but also contain non-overlapping digit classes, contrasting the usual adaptation setup with a shared label space. Our goal is to learn a feature space that separates the digit classes not only within each domain, but also across the two.

We construct a feature generator $f$ composed of a CNN encoder followed by two fully-connected (FC) layers and a feature transfer module $g$ composed of MLP with residual connections. Outputs of $f$ and $g$ are then fed to discriminators $D_1$ and $D_2$ parameterized by MLPs to induce domain adversarial and domain separation losses respectively. We provide more architecture details in Appendix B.1.

We visualize t-SNE plots of generator features in Figure 2. Without an adaptation (Figure 2(a)), features of digits from the target domain are heavily mixed with those from the source domain as

| Model | Verification | | | | Identification | | | |
|---|---|---|---|---|---|---|---|---|
| | CAU | AA | EA | ALL | CAU | AA | EA | ALL |
| $\text{Sup}^C$ | 98.39 | 92.24 | 93.41 | 95.58 | 90.07 | 69.64 | 76.37 | 77.97 |
| $\text{Sup}^{C,A,E}$ | **98.43** | **97.16** | 97.05 | **98.15** | 90.16 | **84.02** | **84.38** | **85.75** |
| $\text{DANN}\backslash\mathcal{L}_{\text{recon}}$ | 98.36 | 94.54 | 95.02 | 96.84 | 90.01 | 73.05 | 74.94 | 77.99 |
| DANN | 98.36 | 95.37 | 96.36 | 97.34 | 90.34 | 74.88 | 79.39 | 79.83 |
| FTN | 98.36 | 95.62 | 96.64 | 97.68 | 90.54 | 75.35 | 80.69 | 81.28 |
| DANN+MCEM | 98.39 | 96.36 | 97.34 | 97.88 | 90.77 | 80.30 | 83.07 | 82.69 |
| FTN+MCEM | 98.37 | 96.76 | **97.40** | 98.08 | **90.95** | 80.75 | 83.71 | 84.16 |

**Table 1:** Verification and identification accuracy on the Cross Ethnicity Faces (CEF) dataset. For supervised models, we report results trained on labeled CAU ($\text{Sup}^C$) or on labeled CAU, AA, EA domains ($\text{Sup}^{C,A,E}$); for adaptation, we evaluate DANN and FTN, without and with multi-class entropy minimization (MCEM).

| Model | CAU vs. AA, EA | AA vs. CAU | EA vs. CAU |
|---|---|---|---|
| $\text{Sup}^C$ | 91.67 | 95.42 | 94.87 |
| DANN | 89.91 | 84.78 | 91.47 |
| FTN | 92.29 | 88.09 | 92.07 |

**Table 2:** Cross domain identification accuracy on CEF, with CAU evaluated against AA + EA combined, AA against CAU and EA against CAU.

well as one another. The model reaches $1.3\%$ verification error in the source domain but as high as $27.3\%$ in the target domain. Though DANN in Figure 2(b) shows better separation with a reduced target verification error of $2.2\%$, there still exists significant overlap between digit classes across two domains, such as 3/5, 4/9, 0/6 and 2/8. As a result, a domain classifier trained to distinguish source and target on top of generator features can only attain $11.5\%$ classification error. In contrast, the proposed FTN in Figure 2(c) shows 10 clean clusters without any visual overlap among 10 digits classes from either source or target domain, implying that it not only separates digits within the target domain ($2.1\%$ verification error), but also differentiates them across domains ($0.3\%$ domain classification error).

## 6.2 Cross Ethnicity Face Verification and Recognition

The performances of face recognition engines have significantly improved thanks to recent advances in deep learning for image recognition (Krizhevsky et al., 2012; Simonyan & Zisserman, 2015; Szegedy et al., 2015; He et al., 2016) and publicly available large-scale face recognition datasets (Yi et al., 2014; Guo et al., 2016). However, most public datasets are collected from the web by querying celebrities, with significant label bias towards Caucasian ethnicity. For example, more than $85\%$ of identities are Caucasian for CASIA Web face dataset (Yi et al., 2014). Similarly, $82\%$ are Caucasian (CAU) for MS-Celeb-1M (MS-1M) dataset (Guo et al., 2016), while there are only $9.7\%$ African-American (AA), $6.4\%$ East-Asian (EA) and less than $2\%$ Latino and South-Asian combined.[4]

Such imbalance across ethnicity in labeled training data can result in significant drop in identification performance on data-scarce minorities: the second row of Table 1 shows a model trained on Caucasian dominated dataset performs poorly on the other ethnicities. As expected, if the training data is composed of only Caucasian identities as source domain, the performance over the target domains consisting of the other ethnicities further deteriorates (see row 1 of Table 1). Provided the available labeled source domain contains only Caucasian identities, we subsequently demonstrate that our method can effectively leverage unlabeled data from the non-Caucasian target ethnicity to substantially improve their face verification performances.

**Experimental Setup.** We perform an adaptation from CAU to a mixture of AA and EA. Our experiments use the MS-1M dataset. We first remove identities that both appear in the training and testing sets. The resulting training set consists of $4.04M$ images from $60K$ CAU identities, $398K$ images from $7K$ AA identities, and $308K$ images from $4.6K$ EA identities. For domain adaptation experiments, we use labeled CAU images and unlabeled AA, EA images for training. For supervised experiments to obtain performance lower and upper bound, we use labeled CAU images to train $\text{Sup}^C$ and labeled CAU, AA, EA images to train $\text{Sup}^{C,A,E}$.

We adopt a 38-layer ResNet (He et al., 2016) for the feature generation module. Feature transfer module and discriminators are parameterized with MLPs similarly to Section 6.1. We use 4096-pair loss for training, including for the supervised CNNs. It is worth mentioning that our network architecture

---

[4]We ask AMT to organize the ethnicity of face images into five categories, Caucasian, African-American, East-Asian, South-Asian and Latino. Sample annotated images are shown in Appendix E.

and training scheme result in strongly competitive face recognition performance, comparing to other state-of-the-art methods such as FaceNet (Schroff et al., 2015) on YouTube Faces (Wolf et al., 2011) (97.32% (ours) vs 95.12%) and Neural Aggregation Network (Yang et al., 2017) on IJB-A (see row 2 of Table 3). The complete network architecture and training details are provided in Appendix B.2.

**Evaluation.** We report the performance of the baseline and our proposed models on two standard face recognition benchmarks LFW (Huang et al., 2007) and IJB-A (Klare et al., 2015). Note that these datasets also exhibit significant ethnicity bias.[5]

To highlight the effectiveness of the proposed adaptation approach, we construct individual test set for CAU, AA, EA, each of which contains 10 face images from 200 identities. We refer to our testing set as the Cross-Ethnicity Faces (CEF) dataset. We apply two evaluation metrics on CEF dataset, verification accuracy and identification accuracy. For verification, following the standard protocol (Huang et al., 2007), we construct 10 splits, each containing 900 positive and 900 negative pairs, and compute the accuracy on each split using the threshold found from the other 9 splits. For identification, a pair composed of the reference and the query images from the same identity is considered correct if there is no image from different identity that has higher similarity to the reference image than the query image. We evaluate identification accuracy per ethnicity (200-way) as well as across all ethnicities (600-way).

**Results.** The results on CEF are summarized in Table 1. Cross domain identification accuracy is reported in Table 2, where we use AA and EA as negative classes when evaluating accuracy on CAU and vice versa, as a measure to indicate domain discrepancy. Among adaptation models, DANN without feature reconstruction loss (DANN$\setminus\mathcal{L}_{\text{recon}}$) shows unstable training and easily degenerate, which leads to only marginal improvement upon Sup$^C$. Similar trend is observed while training FTN. Therefore, to ensure training stability, we impose $\mathcal{L}_{\text{recon}}$ as a regularization term for all adaptation models. More analysis on the effectiveness of $\mathcal{L}_{\text{recon}}$ is provided in Appendix C.

When testing on AA and EA with model trained on only the labeled source CAU domain (Sup$^C$), we observe significant performance drops in Table 1. Meanwhile, in Table 2, cross domain identification accuracy is much higher than within domain identification accuracy, i.e., 96.14% of AA vs. CAU is much higher than 71.92% of AA identification in Table 1, indicating 1) significant discrepancy between the feature spaces of the source and target domains and 2) lack of discriminative power for within domain verification task on target ethnicity.

Comparing to Sup$^C$, both DANN and FTN show moderate improvement when testing on AA and EA from CEF (Table 1), demonstrating the effectiveness of domain adversarial learning in transferring within domain verification capability from labeled source domain to unlabeled target domain. Despite the improvement, DANN suffers a notable drawback from adversarial objective which attempts to align identities from different domains, resulting a poor cross domain identification accuracy as shown in Table 2. In contrast, the proposed FTN achieves much higher cross domain identification accuracy, demonstrating both within and cross domain discriminative power.

Additionally, in combination with the multi-class entropy minimization (FTN+MCEM), we further boost the verification and identification accuracy over FTN on AA and EA as well as approach the accuracy of Sup$^{C,A,E}$, the performance upper bound. This indicates that the HDBSCAN-based hierarchical clustering provides high quality pseudo-class labels for MCEM to be effective. Indeed, the clustering algorithm achieves F-score as high as 96.31% and 96.34% on AA and EA. We provide more in-depth analysis on the clustering strategy in Appendix D.

Finally, Table 3 reports the performance of face recognition models on standard verification and recognition benchmarks. We observe similar improvements with our proposed distance metric adaptation when only using labeled CAU, i.e., source domain, as training data. Once the task becomes more challenging thus demands more discriminative power, the advantage of our method becomes more evident, such as in the case of open-set recognition and verification at low FAR.

## 7 CONCLUSION

We address the challenge of unsupervised domain adaptation when the source and the target domains have disjoint label spaces by formulating the classification problem into a verification task. We

---

[5]We find that LFW dataset is composed of 84.1% of CAU, 9.4% of AA, and 6.5% of EA. IJB-A dataset is less biased, but still with a dominating 71.6% CAU versus 8.2% AA and 10.6% EA.

| Model | LFW | | | | IJB-A (verification) | | | IJB-A (id.) | |
|---|---|---|---|---|---|---|---|---|---|
| | VRF | CLS | 0.01 | 0.001 | 0.01 | 0.001 | 0.0001 | rank-1 | rank-5 |
| $\text{Sup}^{C}$ | 99.57 | 98.95 | 86.07 | 66.61 | 92.67 | 76.65 | 50.32 | 94.31 | 97.25 |
| $\text{Sup}^{C,A,E}$ | **99.72** | 98.79 | 96.81 | 91.11 | **95.57** | 87.45 | 76.45 | 94.73 | 97.19 |
| $\text{DANN}\backslash\mathcal{L}_{\text{recon}}$ | 99.43 | 98.98 | 96.81 | 91.44 | 94.23 | 86.87 | 73.80 | 94.27 | 97.03 |
| DANN | 99.63 | 98.95 | 97.15 | 93.46 | 95.54 | 88.64 | 77.13 | 94.59 | **97.31** |
| FTN | 99.63 | 99.11 | 97.15 | 92.95 | 95.07 | 88.45 | 77.70 | 94.48 | 97.19 |
| DANN+MCEM | 99.63 | 99.08 | **97.65** | **94.97** | 95.28 | **88.78** | 77.30 | 94.75 | 97.30 |
| FTN+MCEM | 99.65 | **99.14** | 96.98 | 93.46 | 94.63 | 88.28 | **77.98** | **94.79** | 97.00 |

**Table 3:** Face verification and recognition performance on LFW and IJB-A. From left to right, verification (VRF), closed-set (CLS) and open-set recognition at FAR = 0.01 and 0.001 (Best-Rowden et al., 2014) on LFW, and verification at different FAR and identification (id.) at rank-$k$ on IJB-A are reported.

propose a Feature Transfer Network, allowing simultaneous optimization of domain adversarial loss and domain separation loss, as well as a variant of $N$-pair metric loss for entropy minimization on the target domain where the ground-truth label structure is unknown, to further improve the adaptation quality. Our proposed framework excels at both within-domain and cross-domain verification tasks. As an application, we demonstrate cross-ethnicity face verification that overcomes label biases in training data, achieving high accuracy even for unlabeled ethnicity domains, which we believe is a result with vital social significance.

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

# Appendix

## A   DERIVATION FOR GENERALIZATION BOUND OF TARGET DOMAIN VERIFICATION LOSS

Let $(\mathcal{X}, \mathcal{F})$ and $(\mathcal{Z}, \mathcal{G})$ be measurable input and feature spaces respectively and a feature extractor $R : \mathcal{X} \to \mathcal{Z}$ be a measurable function. Let $\mu$ be a probability measure on $\mathcal{X}$ corresponding to the data distribution. Let $(\mathcal{X}_1, \mathcal{F}, \mu_1) = (\mathcal{X}_2, \mathcal{F}, \mu_2) = (\mathcal{X}, \mathcal{F}, \mu)$ and $\mu_{12} = \mu_1 \times \mu_2$ on $\mathcal{X}_1 \times \mathcal{X}_2$ be the unique product measure (Forrest). Similarly, we construct $\mathcal{Z}_1 \times \mathcal{Z}_2$ where $(\mathcal{Z}_1, \mathcal{G}) = (\mathcal{Z}_2, \mathcal{G}) = (\mathcal{Z}, \mathcal{G})$. Since $R$ is measurable, $R^2 : \mathcal{X}_1 \times \mathcal{X}_2 \to \mathcal{Z}_1 \times \mathcal{Z}_2$ where $R^2(x_1, x_2) = (R(x_1), R(x_2))$ is also measurable (see Lemma 1 for the proof). Then we can obtain an induced probability measure for $\mathcal{Z}_1 \times \mathcal{Z}_2$ from $R^2$, denoted as $\tilde{\mu}_{12} = \mu_{12} \circ (R^2)^{-1}$ (Proposition 1.34 from (Lalley, 2017)).

Let $Y : \mathcal{X}_1 \times \mathcal{X}_2 \to \{0, 1\}$, where 1 represents the pair from the same identity and 0 otherwise,[6] be the stochastic target function for ground truth labeling, $\phi(x_1, x_2) = \mathbb{E}[Y(x_1, x_2)]$ be the expectation of the label at $(x_1, x_2)$, and $\tilde{\phi}(z_1, z_2) = \mathbb{E}[\phi(x_1, x_2) | R(x_1) = z_1, R(x_2) = z_2]$ be the conditional expectation of $\phi$ given the value of $R^2(x_1, x_2) = (z_1, z_2)$. Now, consider two domains, namely the source domain with probability measure $\mu^S$ over $\mathcal{X}$ and induced probability measure $\tilde{\mu}^S$ over $\mathcal{Z}_1 \times \mathcal{Z}_2$, as well as target domain counterparts $\mu^T$ and $\tilde{\mu}^T$. Provided with a deterministic hypothesis class $\mathcal{H} \subseteq \{g : \mathcal{Z}_1 \times \mathcal{Z}_2 \to \{0, 1\}\}$ of VC-dimension $d$, suppose there exists a function $h \in \mathcal{H}$ that can predict both source and target domains reasonably well. Then, we can quantify $\tilde{\phi}$ to be $\lambda$-close to $\mathcal{H}$:

$$\inf_{h \in \mathcal{H}} \epsilon_S(h) + \epsilon_T(h) \leq \lambda, \text{ where } \epsilon_i(h) = \int |\tilde{\phi}(z_1, z_2) - h(z_1, z_2)| d\tilde{\mu}^i.$$

We are ready to define the variational distance between the two domains with respect to $\mathcal{H}$:

$$d_{\mathcal{H}}(\tilde{\mu}^S, \tilde{\mu}^T) = 2 \sup_{A \in \mathcal{A}} |\tilde{\mu}^S(A) - \tilde{\mu}^T(A)|, \ \mathcal{A} = \{A_h = \{(z_1, z_2) \in \mathcal{Z}_1 \times \mathcal{Z}_2 : h(z_1, z_2) = 1\}, h \in \mathcal{H}\}.$$

So far, we have successfully prepared the components in our verification setup to meet the assumptions and the format required by Theorem 1 from (Ben-David et al., 2007). We may now directly apply the theorem:

**Theorem 1.** *Randomly sample a labeled set of size $m$ by applying $R^2$ to samples from $\mathcal{X}_1 \times \mathcal{X}_2$ with labels defined according to $Y$, with probability at least $1 - \delta$, $\forall h \in \mathcal{H}$*

$$\epsilon_T(h) \leq \hat{\epsilon}_S(h) + \sqrt{\frac{4}{m}\left(d \log \frac{2m}{d} + d + \log \frac{4}{\delta}\right)} + d_{\mathcal{H}}(\tilde{\mu}^S, \tilde{\mu}^T) + \lambda.$$

Furthermore, $d_{\mathcal{H}}(\tilde{\mu}^S, \tilde{\mu}^T)$ can be empirically approximated by finite samples from both domain (Kifer et al., 2004), using the binary classifier from $\mathcal{H}$ that can best distinguishes pairs of samples between two domains. Following Theorem 2 from (Ben-David et al., 2007), let $\tilde{U}_S$ and $\tilde{U}_T$ consist of $n$ random pairs of samples from source and target each, with probability at least $1 - \delta$, we have :

$$d_{\mathcal{H}}(\tilde{\mu}^S, \tilde{\mu}^T) \leq d_{\mathcal{H}}(\tilde{U}_S, \tilde{U}_T) + \sqrt{\frac{d \log(2n) + \log \frac{4}{\delta}}{n}},$$

where $d_{\mathcal{H}}(\tilde{U}_S, \tilde{U}_T) = 2\left(1 - 2 \min_{h \in \mathcal{H}} \frac{1}{2n} \sum_{i=1}^{2n} |h(z_{1,i}, z_{2,i}) - \mathbf{1}\{(z_{1,i}, z_{2,i}) \in \tilde{U}_S\}|\right)$.

For completeness of our analysis, we formalize and prove in Lemma 1 that $R^2$ is measurable.

**Lemma 1.** *Let $(\mathcal{X}, \mathcal{F}, \mu)$ and $(\mathcal{Z}, \mathcal{G}, \tilde{\mu})$ be measurable spaces and let $(\mathcal{X} \times \mathcal{X}, \sigma(\mathcal{F} \times \mathcal{F}), \mu \times \mu)$, $(\mathcal{Z} \times \mathcal{Z}, \sigma(\mathcal{G} \times \mathcal{G}), \tilde{\mu} \times \tilde{\mu})$ be their product spaces with the product measures. Let $R : \mathcal{X} \to \mathcal{Z}$ be a measurable function, then $R^2 : \mathcal{X} \times \mathcal{X} \to \mathcal{Z} \times \mathcal{Z}$ where $R^2(x_1, x_2) = (R(x_1), R(x_2))$ is also measurable.*

*Proof.* As the $\sigma$-algebra of $\mathcal{Z} \times \mathcal{Z}$ is generated by $\mathcal{G} \times \mathcal{G}$, we only need to show that the pre-image of any generator is measurable. Let $G_1 \times G_2 \in \mathcal{G} \times \mathcal{G}$, then it is easy to see that $(R^2)^{-1}(G_1 \times G_2) = R^{-1}(G_1) \times R^{-1}(G_2)$. Since $R$ is a measurable function, hence $R^{-1}(G_1)$ and $R^{-1}(G_2)$ are measurable and so is $R^{-1}(G_1) \times R^{-1}(G_2)$ measurable. □

---

[6]Rigorously, $Y(x_1, x_2)$ is a Bernoulli random variable with outcome space containing 0 and 1.

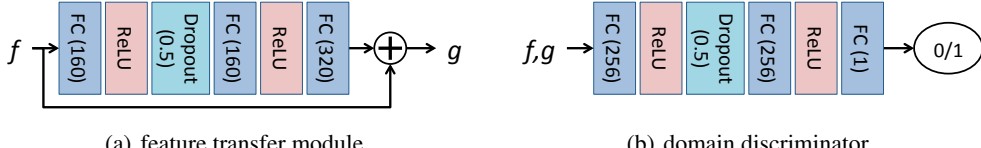

(a) feature transfer module

(b) domain discriminator

**Figure S1:** Network architecture of feature transfer module and domain discriminators.

| operation | kernel | output size |
|---|---|---|
| Conv1-1 + ReLU | 3×3 | 32×32×32 |
| Conv1-2 + ReLU | 3×3 | 32×32×32 |
| max pooling | 2×2 | 16×16×32 |
| Conv2-1 + ReLU | 3×3 | 16×16×64 |
| Conv2-2 + ReLU | 3×3 | 16×16×64 |
| max pooling | 2×2 | 8×8×64 |
| Conv3-1 + ReLU | 3×3 | 8×8×128 |
| Conv3-2 + ReLU | 3×3 | 8×8×128 |
| max pooling | 2×2 | 4×4×128 |
| FC1 + ReLU | – | 128 |
| FC2 | – | 128 |
| Normalize and Scale (2) | – | 128 |

**Table S1:** Network architecture for digit experiments.

# B   NETWORK ARCHITECTURE AND TRAINING DETAILS

## B.1   TOY EXPERIMENTS: MNIST-M $(0-4)$ TO MNIST $(5-9)$

Following (Haeusser et al., 2017), we preprocess the data by subtracting a channel-wise pixel mean and dividing by channel-wise standard deviation of pixel values. For MNIST examples, we also apply color-intensity inversion. All images are resized into 32×32 with 3 channels.

Our feature generator module is composed of 6 convolution layers and 3 max-pooling layers followed by 2 fully-connected layers. We use ReLU (Nair & Hinton, 2010) after convolution layers. The output dimension of the feature generator module is 128 and is normalized to have L2-norm of 2. The full description of the generator module is in Table S1.

The feature transfer module maps 128 dimensional vector into the same dimensional vector using two fully-connected layers $(128-256-256-128)$ and residual connection as in Figure 1(a). Discriminator architectures are similar to that in Figure 1(b) but with fully-connected layers whose output dimensions are 128 instead of 256.

We use Adam stochastic optimizer with learning rate of 0.0003, $\lambda_1 = 0.3$ and $\lambda_2 = 0.03$ to train FTN.

## B.2   CROSS ETHNICITY FACE VERIFICATION AND RECOGNITION

Our experimental protocols, such as data preprocessing and network architecture, closely follow those of (Sohn et al., 2017). We preprocess face images by detecting (Yang et al., 2016), aligning (Yu et al., 2016), and cropping to provide face images of size $110 \times 110$. The data is prepared for network training by random cropping into $100 \times 100$ with horizontal flip with a $50\%$ chance and converting into gray-scale.

Our feature generation module contains 38 layers of convolution with several residual blocks and max pooling layers. We use ReLU (Nair & Hinton, 2010) for most of the layers in combination with maxout nonlinearities (Goodfellow et al., 2013). We add $7 \times 7$ average pooling layer on top of the last convolution layer. The output of the feature generation module is 320 dimensional vector and is normalized to have L2-norm of size 12. The full description of the model is in Table S2.

The feature transfer module maps 320 dimensional output vector from feature generation module into the same dimensional vector using two fully-connected layers and residual connection. The architecture of feature transfer module is described in Figure 1(a). Discriminators have similar network architecture besides different numbers of neurons and omitted residual connection.

All models, including supervised CNNs (Sup$^C$, Sup$^{C,A,E}$), are trained with 4096-pair loss. For Sup$^C$ and Sup$^{C,A,E}$, we use Adam stochastic optimizer (Kingma & Ba, 2015) with the learning rate of $0.0003$ for the first $12K$ updates and $0.0001$ and $0.00003$ for the next two subsequent $3K$ updates.

Our feature generation module is initialized with the Sup$^C$ model, which is also used as a reference network for feature reconstruction loss as described in Section 4.3. Other modules of our model, such as feature generation module and discriminators, are initialized randomly. All modules are then updated with the learning rate of $0.00003$. Hyperparameters of different models are summarized in Table S3.

| operation | kernel | output size |
|---|---|---|
| Conv1-1 + ReLU | $3\times3$ | $100\times100\times32$ |
| Conv1-2 + Maxout (2) | $3\times3$ | $100\times100\times64$ |
| max pooling | $2\times2$ | $50\times50\times64$ |
| ResBlock + ReLU $\times2$ | $3\times3, 64-64-64$ | $50\times50\times64$ |
| Conv2 + Maxout (2) | $3\times3$ | $50\times50\times128$ |
| max pooling | $2\times2$ | $25\times25\times128$ |
| ResBlock + ReLU $\times4$ | $3\times3, 128-96-128$ | $25\times25\times128$ |
| Conv3 + Maxout (2) | $3\times3$ | $25\times25\times192$ |
| max pooling | $2\times2$ | $13\times13\times192$ |
| ResBlock + ReLU $\times8$ | $3\times3, 192-128-192$ | $13\times13\times192$ |
| Conv4 + Maxout (2) | $3\times3$ | $13\times13\times256$ |
| max pooling | $2\times2$ | $7\times7\times256$ |
| ResBlock + ReLU $\times2$ | $3\times3, 256-160-256$ | $7\times7\times256$ |
| Conv5 + Maxout (2) | $3\times3$ | $7\times7\times320$ |
| avg pooling | $7\times7$ | $1\times1\times320$ |
| Normalize and Scale (12) | $-$ | $320$ |

**Table S2:** Network architecture for face experiments.

| | $\lambda_1$ | $\lambda_2$ | $\lambda_3$ | $\lambda_4$ |
|---|---|---|---|---|
| DANN | 0.1 | $-$ | 0.1 | 0.01 |
| FTN | 0.03 | 0.1 | 0.03 | 0.01 |
| FTN+MCEM | 0.03 | 0.1 | 0.03 | 0.003 |

**Table S3:** Optimal hyperparameter settings of different adaptation models.

## C   IMPACT OF FEATURE RECONSTRUCTION LOSS ON DOMAIN ADVERSARIAL TRAINING

We demonstrate the effectiveness of feature reconstruction loss in stabilizing the domain adversarial training in DANN framework. We train four different DANN models with different configurations of $\lambda_3$ and $\lambda_4$. We visualize in Figure S2 the performance curves of identification accuracy evaluated on the AA, EA, and CAU ethnicities of CEF dataset. Note that we stop training early on when the performance start to degrade significantly. Therefore, $x$-axis, the number of training epoch, of different curves are different. $y$-axis represents the identification accuracy.

As we see in Figure S2, the performance of all models on the target ethnicities start to improve in the beginning of training from those of the pretrained reference network. Soon after, however, the accuracy starts to drop when values of either $\lambda_3$ or $\lambda_4$ are set to $0$. Note that even in that situation the performance on the CAU set still remains high, which implies the failure of discriminative

information transfer. On the other hand, our proposed feature reconstruction loss with non-zero values of $\lambda_3$ and $\lambda_4$ (Figure 2(d)) shows much more stable performance curve. Nonetheless, values of $\lambda_3$ and $\lambda_4$ should be carefully selected since the feature generation module of DANNs or FTNs will remain almost the same to the reference network when they are set too strong and the effectiveness of the domain adversarial loss will be reduced. In our experiment, we use $\lambda_3 = 0.1$ and $\lambda_4 = 0.01$ for DANN, $\lambda_3 = 0.03$ and $\lambda_4 = 0.01$ for FTN. For FTN with entropy minimization we further reduce $\lambda_4 = 0.003$ to give more flexibility in updating model parameters based on entropy loss.

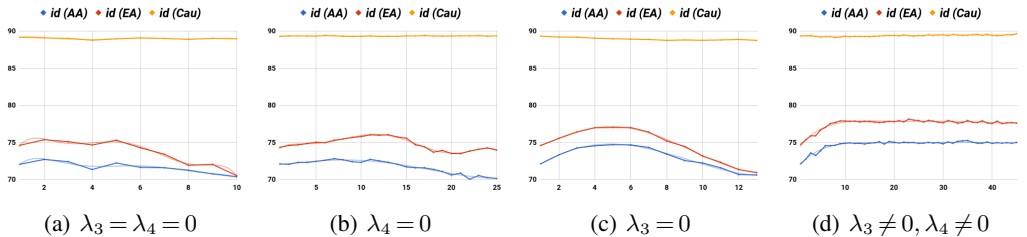

(a) $\lambda_3 = \lambda_4 = 0$      (b) $\lambda_4 = 0$      (c) $\lambda_3 = 0$      (d) $\lambda_3 \neq 0, \lambda_4 \neq 0$

**Figure S2:** Performance curves of identification accuracy per ethnicity subset on the CEF datasets. The accuracy of DANNs with different values of $\lambda_3$ for $\lambda_4$ are visualized.

## D    PERFORMANCE OF UNSUPERVISED HIERARCHICAL CLUSTERING

In this section, we provide analysis on the performance of our clustering strategy by measuring the clustering accuracy. Specifically, we measure the verification precision and recall as follows:

$$\text{Precision} = \frac{\sum_{x_1, x_2 \in \mathcal{X}_T} 1\{y_1 = y_2, \hat{y}_1 = \hat{y}_2\}}{\sum_{x_1, x_2 \in \mathcal{X}_T} 1\{\hat{y}_1 = \hat{y}_2\}}, \text{ Recall} = \frac{\sum_{x_1, x_2 \in \mathcal{X}_T} 1\{y_1 = y_2, \hat{y}_1 = \hat{y}_2\}}{\sum_{x_1, x_2 \in \mathcal{X}_T} 1\{y_1 = y_2\}} \quad \text{(S1)}$$

where $y_i$ is the ground-truth class label of an example $x_i$, and $\hat{y}_i$ is an index of an assigned cluster. Precision computes the proportion of positive pairs among pairs assigned to the same cluster, i.e., purity of returned clusters, and recall computes the proportion of positive pairs assigned to the same cluster. Ideally, we expect high precision and high recall, i.e., high F-score, to ensure examples with the same class labels are assigned to the same cluster. Note that we only use clusters of size 5 or larger as new target classes and discard examples assigned to a cluster whose size is less than 5.

Here, in addition to our proposed clustering strategy, we also evaluate the clustering performance that clusters target examples by finding a nearest classes or examples from the source domain, which are shown to be effective for zero-shot learning (Vinyals et al., 2016) or semi-supervised domain adaptation with disjoint source and target classes (Luo et al., 2017). In this case, we call two examples from the target domain are assigned to the same cluster if the nearest source examples are the same. We also measure the clustering performance by matching the nearest source classes.

The summary result is provided in Table S4. Firstly, we observe extremely low precision when using source domain examples or clusters as a proxy to relate target examples. We believe that this idea of "clustering by finding the nearest source classes" works under a cross-category similarity assumption between disjoint classes of source and target domains. In other words, it assumes that there exists a certain source class closer to examples from certain target class, so that those examples from the same target class can be clustered around that source class, even though those matching source and target classes are indeed different (e.g., 3/5, 2/8, 4/9, and 0/6 in Section 6.1). Unfortunately, such an assumption does not hold for our problem, maybe due to the huge number of identity classes ($60K$) in the source domain.

On the other hand, using hierarchical clustering on target features achieves significantly higher precision and recall. Especially, when using embedding vectors of $\text{Sup}^C$, we achieve $100\%$ precision, which means that all clusters are pure even though some ground-truth classes might be separated into multiple clusters. We observe slightly lower precision using FTN features but much higher recall, achieving higher F-score overall. Further, the number of examples returned with FTN feature ($253K$ and $195K$ for AA and EA, respectively) is higher than with $\text{Sup}^C$ feature ($217K$ and $165K$). Repeating the process using feature of FTN+MCEM model improves the F-score while returning more target examples that are with cluster assignment ($276K$ and $214K$). This not only shows the

|  | source example | | source center | | HDBSCAN | | FTN | | FTN+MCEM | |
|---|---|---|---|---|---|---|---|---|---|---|
|  | AA | EA | AA | EA | AA | EA | AA | EA | AA | EA |
| Precision | 0.11 | 0.12 | 0.30 | 0.08 | **100** | **100** | 95.36 | 96.23 | 95.79 | 96.10 |
| Recall | 25.64 | 31.11 | 22.54 | 48.60 | 88.66 | 81.74 | 97.29 | 96.46 | **97.81** | **96.89** |
| F-score | 0.22 | 0.25 | 0.58 | 0.16 | 93.99 | 89.95 | 96.31 | 96.34 | **96.79** | **96.49** |

**Table S4:** Verification precision and recall of clustering methods, such as projection to source example or source class center, or hierarchical clustering using embeddings of Sup$^C$ (HDBSCAN) or our proposed FTN model. Furthermore, we repeat the clustering using the FTN with multi-class entropy minimization model (FTN+MCEM) and report the clustering accuracy.

improved discriminative quality of features by FTNs, but also suggests a potential tool for automatic labeling of unlabeled data by iterative training of FTN model and hierarchical clustering.

# E   VISUALIZATION OF ETHNICITY ANNOTATED IMAGE SAMPLES

We visualize few images from each ethnicity subset in Figure S3 for annotation quality assurance.

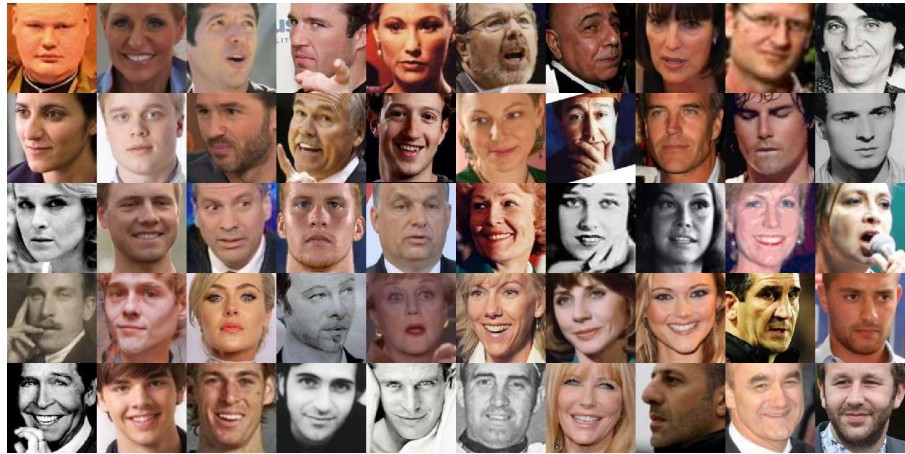

(a) Caucasian

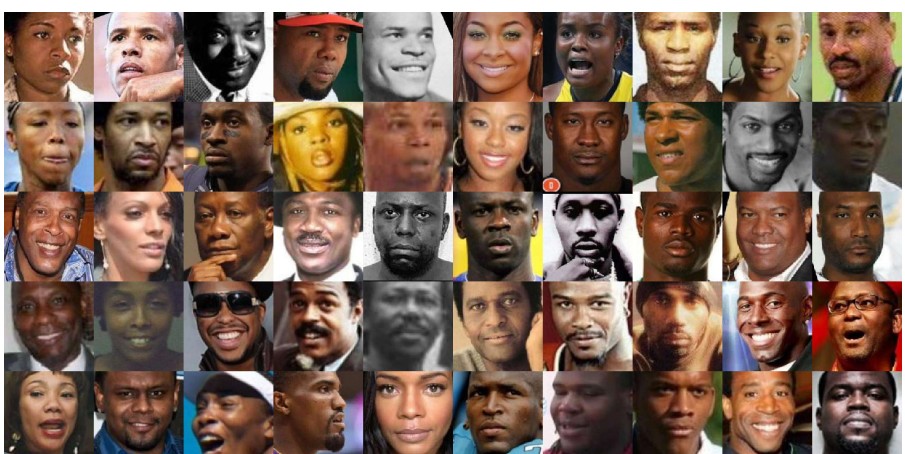

(b) African-American

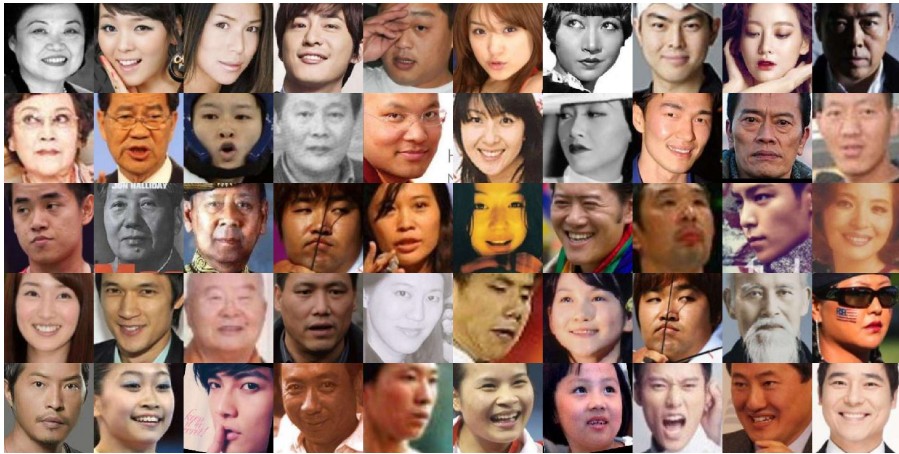

(c) East-Asian

**Figure S3:** Face images of Caucasian, African-American, and East-Asian sampled from MS-1M dataset.

