# OpenReview forum: "Unsupervised Domain Adaptation for Distance Metric Learning"
_ICLR.cc/2019/Conference_

### Official Review · AnonReviewer2 · 2018-10-29
**Interesting paper addressing a difficult problem. Good formalization and reasonable evaluation**

**Rating:** 8
**Confidence:** 4

**Review:**

I like the idea of the paper and I believe it addressing a very relevant problem. While the authors provide a good formalization of the problem and convincing demonstration of the generalization bound, the evaluation could have been better by including some more challenging experiments to really prove the point of the paper. It is surely good to present the toy example with the MNIST dataset but the ethnicity domain is less difficult than what the authors claim. This is also pretty evident from the results presented (e.g., in Table 3). The proposed approach provides maybe slightly better results than the state of the art but the results do not seem to be statistically significant. This is probable also due to the fact that the problem itself is made simpler by the cropped faces, no background, etc. I would have preferred to see an application domain where the improvement would be more substantial. Nevertheless, I think the theoretical presentation is good and I believe the manuscript has very good potential.

---

> ### Author Response · Authors · 2018-11-21
> **Response to AnonReviewer2**
>
> We thank the reviewer for their valuable comments.
>
> We understand the concern in Table 3 that the performance improvement is not as significant as in Table 1. As mentioned in footnote 5, we observe that the ethnicity bias not only exists in the training dataset, but also in public benchmark datasets, such as LFW or IJB-A. While we observe the benefit of FTN over source only model in all evaluation metrics or over DANN in low FAR regime, thus requiring more within as well as cross-domain discriminativeness, we believe that these datasets may not be the best to evaluate the fairness of face recognition algorithms. This indeed is our motivation to collect an ethnicity-balanced test dataset for fair evaluation. We will make the dataset publicly available to the community upon publication.

---

### Official Review · AnonReviewer3 · 2018-11-02
**The motivation is clear but the experiments are not sufficient.**

**Rating:** 5
**Confidence:** 4

**Review:**

In this work, authors consider transfer learning problem when labels for the target domain is not available. Unlike the conventional transfer learning, they introduce a new loss that separates examples from different domains. Besides, they apply the multi-class entropy minimization to optimize the performance in the target domain. Here are my concerns.
1.	The concept is not clear. For domain adaptation, we usually assume domains share the same label space. When labels are different, it can be a transfer learning problem.
2.	Optimizing the verification loss is conventional for distance metric learning based transfer learning and authors should discuss more in the related work.
3.	The empirical study is not sufficient. There lacks the method of transfer learning with distance metric learning. Moreover, the major improvement seems from the MCEM rather than the proposed network. How about DANN+MCEM?

---

> ### Author Response · Authors · 2018-11-21
> **Response to AnonReviewer3**
>
> We thank the reviewer for their valuable comments.
>
> (In response to 3) We argue that many distance metric adaptation or transfer learning algorithms in deep learning are based on distribution matching. For example, [3,4] uses discriminator-based adversarial loss and [5] uses kernel-based MMD loss to reduce the domain discrepancy. Regardless of the discriminator or the kernel, these methods will push two domains closeby and thus have the same limitation as DANN. The proposed FTN resolves this issue by learning “domain-equivariant” representation and we provide empirical evidence (e.g. Table 2 or Figure 2(b-c)) using DANN as the most representative baseline. While one may try adding more components, such as deep supervision (e.g., applying MMD loss at multiple feature layers) as in [5], we believe that our contribution is orthogonal and complementary to those additional components.
>
> (In response to 3) We note that the MCEM is one of our novel contributions, which is only made available through our view on converting the classification task into verification. We agree that it plays a critical role to obtain a highly discriminative representation. For example, [6] considers a similar setting of domain adaptation with disjoint label spaces but they require labeled examples and complete definition of the label space of the target domain to apply classification-based adversarial adaptation learning and entropy regularization. Nonetheless, we provide the within-domain (Table 1) and cross-domain (Table 2) identification accuracy of DANN+MCEM below. We will include this result in the revision:
>
> DANN (for within-domain identification, CAU / AA / EA / ALL; for cross-domain, CAU / AA / EA):
> within-domain identification: 89.5 / 75.3 / 78.0 / 78.9
> cross-domain identification: 89.6 / 83.9 / 86.5
>
> DANN+MCEM:
> within-domain identification: 90.0 / 80.1 / 81.4 / 81.9
> cross-domain identification: 89.4 / 87.1 / 89.1
>
> FTN+MCEM:
> within-domain identification: 90.3 / 80.7 / 82.3 / 83.4
> cross-domain identification: 94.0 / 93.1 / 92.8
>
> Similarly to the FTN, we observe improvement using MCEM with DANN, as compared to the DANN only model. Comparing between adaptation models with MCEM, we still observe better performance when combined with FTN. Especially, the contrast in performance becomes significant in cross-domain identification task, which confirms the unique capability of FTN in learning to transfer discriminative knowledge by alignment while separating representations across domains.
>
>
> (In response to 1) Our problem setting is adaptation from labeled source to unlabeled target with disjoint label spaces. Following the nomenclature of [1], it contains flavors from both domain adaptation (DA) and transfer learning (TL). The difference in input distribution between source and target domains and the lack of labels in the target domain are similar to that of DA or transductive TL [1], while the difference in label distribution and task definitions between two domains is akin to inductive TL [1,2]. In our work, we formalize this problem in domain adaptation framework using verification as a common task. This is a key contribution that allows theoretical analysis on the generalization bound as presented in Section 3 and Appendix A, while also allowing important novel applications like cross-ethnicity face recognition.
>
>
> (In response to 2) We acknowledged in the second paragraph of Section 2 some existing works on domain adaptation that use the verification loss for problems such as face recognition and person re-identification, while highlighting our novel contribution. We will include more discussion and references [5] related to this.
>
>
> [1] Pan and Yang, A survey on Transfer Learning, 2010
> [2] Daume, https://nlpers.blogspot.com/2007/11/domain-adaptation-vs-transfer-learning.html
> [3] Ganin et al., Domain Adversarial Training of Neural Networks, JMLR 2016
> [4] Sohn et al., Unsupervised domain adaptation for face recognition in unlabeled videos, ICCV 2017
> [5] Hu et al., Deep Transfer Metric Learning, CVPR 2015
> [6] Luo et al., Label efficient learning of transferable representations across domains and tasks, NIPS 2017

---

### Official Review · AnonReviewer1 · 2018-11-02
**A good paper addressing domain adaptation for disjoint labels.**

**Rating:** 8
**Confidence:** 5

**Review:**

The authors studied an interesting problem of unsupervised domain adaptation when the source and the target domains have disjoin labels spaces. The paper proposed a novel feature transfer network, that optimizes domain adversarial loss and domain separation loss.

Strengths:

1) The proposed approach on Feature Transfer Network was novel and interesting.
2) The paper was very well written with a good analysis of various choices.
3) Extensive empirical analysis on multi-class settings with a traditional MNIST dataset and a real-world face recognition dataset.


Weakness:
1) Practical considerations addressing feature reconstruction loss needs more explanation.

Comments:

The technical contribution of the paper was sound and novel. The paper considered existing work and in a good way generalizes and extends into disjoint label spaces. It was easy to read and follow, most parts of the paper including the Appendix make it a good contribution. However, the reviewer has the following suggestions"

1. Under the practical considerations for preventing the mode collapse via feature reconstruction, how is the reference network trained? In the Equation(6) for feature reconstruction, the f_ref term maps the source and target domain examples to new feature space. What do you mean by references network trained on the label data? Please clarify.

2. Under the practical considerations for replacing the verification loss, it is said that "Our theoretical analysis suggests to use a verification
the loss that compares the similarity between a pair of images" - Can you please cite the references to make it easier for the reader to follow.

---

> ### Author Response · Authors · 2018-11-21
> **clarification on feature reconstruction loss**
>
> We thank the reviewer for their valuable comments.
>
> 1. We clarify that the reference network is pretrained on the labeled source data and fixed over the training of DANN/FTN. In other words, the gradient in Equation(6) is only backpropagated through f, but not through f_{ref}.
>
> We note that the training procedure of reference network resembles the training of teacher network in distillation framework [1], in the sense that both teacher network and our reference network are “pretrained and fixed” during the training of student or DANN/FTN, respectively.
>
> [1] Hinton et al., Distilling the knowledge in a neural network, NIPS 2014 DL Workshop
>
> 2. We will add a reference (section 3 and appendix) as suggested.

---

### Public Comment · ~Hui-Po_Wang1 · 2018-12-06
**What's the main task you want to address**

Hi authors,

I appreciate you provide thorough and various extension of existing loss functions. However, I would like to know further what's the main problem you want to solve in this work. It seems not to be clear to me.

Let me make a guess and maybe explain the main idea in other words. The proposed method is trying to leverage the "semantic" knowledge in the source domain and perform "clustering" on those target samples with unseen labels (because labels are disjoint).

Assuming I am correct above, I would like to ask the following questions:

In conventional domain adaptation problem, we usually assume that both domains share some common knowledge so that you can utilize the knowledge (labels and corresponding discriminative power) from the source domain to solve similar problems in the target domain. In your work, however, both input and label spaces are "disjoint". I am curious what kind of knowledge you would like to transfer to the target domain and how you can make sure that the knowledge can be applied to those target samples with unseen labels. If these problems are not clarified, as mentioned by reviewer 3, I would say the major improvement all comes from MCEM, which performs clustering algorithm on target samples, instead of the proposed method.

If I made any mistake above, please correct me directly.
Thank you for your patient reading.

best,

---

> ### Author Response · Authors · 2018-12-06
> **response**
>
> Hi Hui-Po,
>
> Thanks for your comment.
>
> As you mentioned, the conventional domain adaptation problems assume the same "task" between the source and the target domains and this allows to transfer discriminative knowledge (e.g., classifier) learned from the source domain to the target domain. On the other hand, not all domains with significant domain shift in the input data space share the same output label spaces, such as cross-ethnicity face recognition or other applications in [1].
>
> In this work, we resolve such limitation of conventional domain adaptation methods and provide a framework that is also applicable when label spaces of two domains are disjoint by converting disjoint identification tasks into a shared verification task. Note that, as we clarified in our response to R3, the conversion of identification to verification allows the problem definition fits perfectly into that of domain adaptation as the source and target domains now have the shared verification task. That being said, the knowledge we are transferring from source to the target domain is verification, i.e., binary classification for pair of data being the same class or not. This is also evident from our theoretical analysis presented in Section 3 and Appendix A where we prove that the verification error defined on the pair of data from the target domain can be bounded by the verification error on the source pair and the domain discrepancy.
>
> Hope this clarifies your concern on "what kind of knowledge is being transferred" between two domains. Please let us know if further clarification is required.
>
> [1] Luo et al., Label efficient learning of transferable representations across domains and tasks, NIPS 2017

---

### Meta-Review · Area_Chair1 · 2018-12-11
**An interesting approach for joint domain adaptation and transfer learning**

**Confidence:** 4
**Recommendation:** Accept (Poster)

**Metareview:**

This paper proposes a new solution for tackling domain adaptation across disjoint label spaces. Two of the reviewers agree that the main technical approach is interesting and novel. The final reviewer asked for clarification of the problem setting which the authors have provided in their rebuttal. We encourage the authors to include this in the final version. However, there is also a consensus that more experimental evaluation would improve the manuscript and complete experimental details are needed for reliable reproduction.